# Intelligent Vehicle Trajectory Tracking Control Based on VFF-RLS Road Friction Coefficient Estimation

Yanxin Nie [1,2] , Yiding Hua [2,*], Minglu Zhang [1] and Xiaojun Zhang [1]

1   School of Mechanical Engineering, Hebei University of Technology, Tianjin 300401, China
2   China Automotive Technology & Research Center Co., Ltd., Tianjin 300300, China
*   Correspondence: huayiding@catarc.ac.cn

**Abstract:** This paper proposes an autonomous vehicle trajectory tracking system that fully considers road friction. When an intelligent vehicle drives at high speed on roads with different friction coefficients, the difficulty of its trajectory tracking control lies in the fast and accurate identification of road friction coefficients. Therefore, an improved strategy is designed based on traditional recursive least squares (RLS), which is utilized for accurate identification of the friction coefficient. First, the tire force and slip rate required for the estimation of the road friction coefficient by constructing the vehicle dynamics model and tire effective model are calculated. In this paper, a variable forgetting factor recursive least squares (VFF-RLS) method is proposed for the construction of the friction coefficient estimator. Second, the identified results are output to the model predictive controller (MPC) constructed in this paper as a way to improve tire slip angle constraints, to realize the trajectory tracking of the intelligent vehicle. Finally, the joint simulation test results of Carsim and Matlab/Simulink show that the trajectory tracking system based on the VFF-RLS friction coefficient estimator has outstanding tracking performance.

**Keywords:** intelligent vehicle; trajectory tracking; friction coefficient estimation; recursive least squares; model predictive control





## 1. Introduction

### 1.1. Rationale

At present, intelligent connected vehicles are generally considered to be future-oriented transportation vehicleriers and are utilized in various complex transportation scenarios [1–3]. The advantages of autonomous driving are super-visual perception ability, accurate trajectory planning decision and precise vehicle control [4–6]. As the core of intelligent vehicle motion control, trajectory tracking control has become the focus of the current field. Related research scholars have vehicleried out a large number of in-depth research studies in the direction of intelligent vehicle trajectory tracking control [7,8].

Trajectory tracking control of the vehicle aims to combine the actual deviation between the actual vehicle location and ideal trajectory, and minimize the lateral deviation and heading deviation through steering control, to achieve dynamic tracking of the referenced trajectory [9,10]. In order to guarantee the driving safety and comfort in trajectory tracking control, while taking into account the high precision and stability requirements of trajectory tracking control, Kim E et al. [11] applied the model predictive control method to propose a trajectory tracking method, to achieve accurate and smooth tracking. Behrooz M et al. [12] designed a trajectory tracking method combining lateral control and yaw moment control, and established the lateral dynamics equation of the trajectory follower vehicle. Simultaneously, in order to guarantee the stability of the vehicle under the maneuvering limit while realizing the lateral trajectory tracking deviation control, Kapania NR et al. [13] proposed a steering controller with feedforward and feedback functions, considering the nonlinear vehicle maneuvering graph-based steering controller. Brown M et al. [14] proposed

a variable control framework that combines trajectory planning and trajectory tracking using MPC. Li BY et al. [15] put forward a trajectory control method for distributed driven electric vehicles based on the potential field method, which does not need to strictly follow the expected trajectory, but forms a feasible region with the expected tracking deviation tolerance limit. Lin F et al. [16] proposed a trajectory tracking control method with vehicle yaw stability control.

### 1.2. State of the Art and Related Work

At present, the research focus of the trajectory tracking control problem is generally to evaluate and analyze the algorithm under good road conditions. The above research ignores the important influence of the road friction coefficient on vehicle stability, and it is difficult to accurately reflect the intelligent vehicle trajectory tracking control system under extreme conditions [17]. Relevant studies have comprehensively considered trajectory tracking and pavement adhesion coefficient estimation. Cui QJ et al. [18] constructed an unscented Kalman to estimate the road friction coefficient, and utilized the estimated result in trajectory tracking control. The researchers in [19] proposed an adaptive estimator for the adhesion coefficient estimation, and used the results to the trajectory tracking control system. Additionally, [20] proposed a tracking control strategy for autonomous vehicles suitable for the sudden change of maximum road friction coefficient. In addition, [21] proposed an external disturbance and road adhesion coefficient estimator for the trajectory tracking control of a front-drive hovercraft. Further, [22] proposed a high-order sliding mode differentiator to reckon the lateral friction coefficient, and based on the above estimation results, a controller using a high-order sliding mode was built to track the referenced trajectory. Choi M et al. [23] constructed a road adhesion coefficient estimator based on the linear RLS and verified the effectiveness of the constructed estimator by means of simulation experiments.

Accurate and reliable road adhesion estimation results are crucial to the performance of the vehicle control system. Therefore, improving the estimation accuracy is an important way to enhance the performance of the control system. At the same time, for the model predictive controller applied to vehicles, the accuracy and real-time ability of input information are particularly important [24,25]. The RLS algorithm is widely used in estimating the road friction coefficient, however, the RLS also has its shortcomings. When the dimension of the matrix increases, the calculation amount of the matrix inversion operation is too large, and the calculation efficiency is low [26]. Therefore, some studies have introduced a forgetting factor based on the RLS algorithm, multiplying the old data by the forgetting factor to cut back the amount of information provided by the old data [27]. However, the forgetting factor is a constant between 0 and 1, that is, attenuating the role of past observations at a stable rate. If the forgetting factor is close to 1, the algorithm is highly accurate, but the tracking ability of the parameters is reduced. Lowering the forgetting factor can improve the tracking ability, but at the same time, reduce the steady-state accuracy [28]. In order to meet these two contradictory needs, this paper proposes an online parameter identification algorithm based on VFF-RLS for the estimation of the road friction coefficient, which can accurately and quickly track the change of friction coefficient under dynamic conditions.

### 1.3. Contributions

The purpose of this paper is to enable intelligent vehicles to quickly and accurately estimate the friction coefficient, and to perform effective trajectory tracking control based on the identification results. Therefore, this paper constructs two main functional modules; the first module is a friction coefficient estimator based on VFF-RLS, and the second module is a trajectory tracking control system based on MPC. The specific structural block diagram is shown in Figure 1.

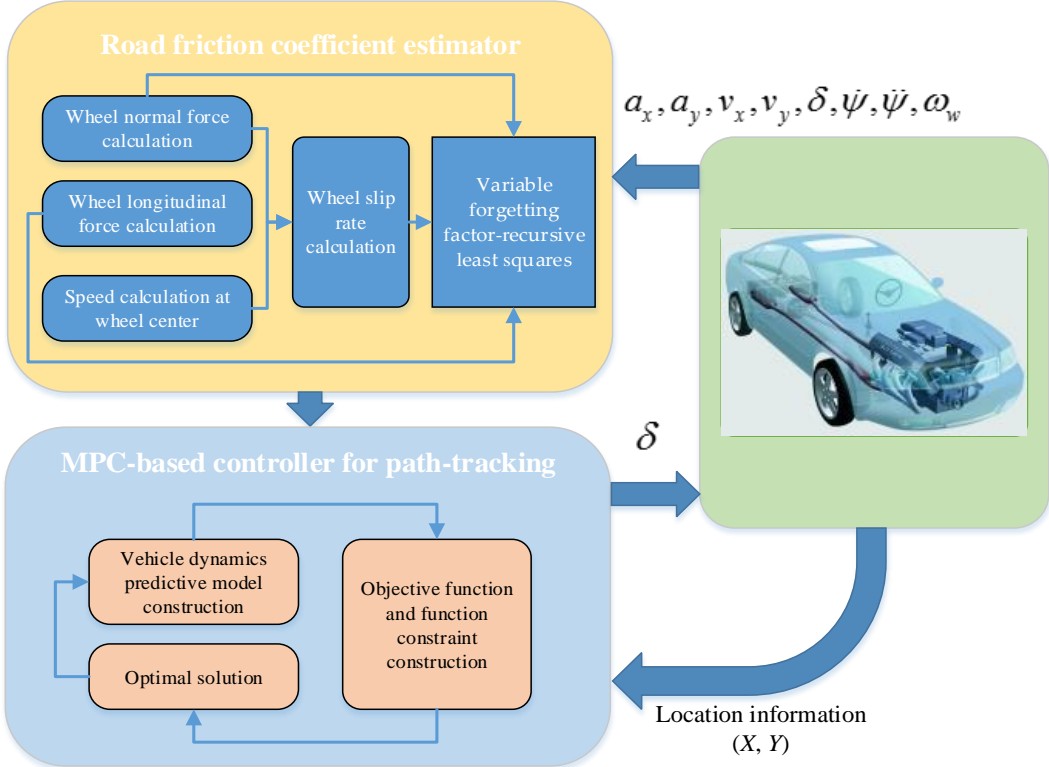

**Figure 1.** The framework of the system.

The remainder of this paper is designed as follows. The second part builds the vehicle dynamics model, the tire effective model and the calculation of related physical quantities. The third part constructs a friction coefficient estimator based on VFF-RLS. The fourth part designs the tracking controller with MPC. The fifth part analyzes the validity of the friction coefficient estimation strategy and the trajectory tracking control system constructed in this paper under the simulation environment. The sixth part gives the conclusion and outlook of the full text.

## 2. Intelligent Vehicle Dynamics Model

### 2.1. Establishment of Vehicle Dynamics Model

A seven-degree-of-freedom vehicle model considering the longitudinal dynamics of the vehicle is established, as shown in Figure 2. The center of mass of the vehicle is taken as the origin of the coordinate system. The longitudinal axis of the vehicle is the *x*-axis, and the forward direction is specified as positive. The center of mass of the vehicle is left and vertical when the longitudinal axis of the vehicle is the *y*-axis, and it is stipulated that the left direction is positive when the vehicle is moving forward. At the same time, the counterclockwise direction of the moment in horizontal plane is defined as positive. The model also makes the following assumptions: (1) The suspension dynamics are ignored. (2) The pitch and roll movements of the vehicle are ignored. (3) Ignore the influence of aerodynamics. (4) The physical properties of each tire are the same.

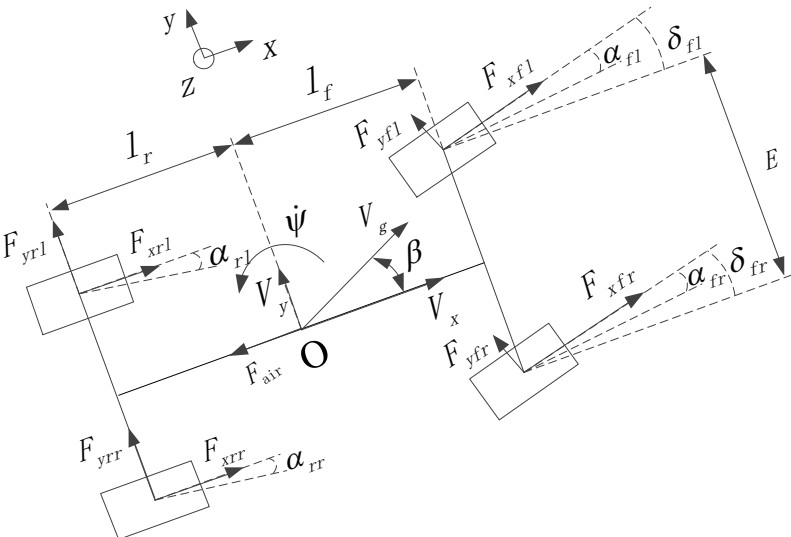

**Figure 2.** Schematic diagram of the seven-degree-of-freedom dynamic model of a vehicle.

According to Newton's second law, the longitudinal kinematics equation can be obtained at the center mass of the vehicle.

When the vehicle is accelerating:

$$ma_x = -F_{yf}\sin\delta + F_{xf}\cos\delta - \frac{1}{2}\rho AC_D u_r^2 - F_{zfl}f_{fl} - F_{zfr}f_{fr} - F_{zrl}f_{zrl} - F_{zrr}f_{zrr} \quad (1)$$

When the vehicle is braking:

$$-ma_x = -F_{xf}\cos\delta - F_{xr} - F_{yf}\sin\delta - \frac{1}{2}\rho AC_D u_r^2 - F_{zfl}f_{fl} - F_{zfr}f_{fr} - F_{zrl}f_{zrl} - F_{zrr}f_{zrr} \quad (2)$$

$$F_{xf}/F_{xr} = 200 : 75 \quad (3)$$

In Figure 2 and Equation (1), $m$ is total vehicle mass, $l_f$ and $l_r$ are the distance from the center of mass to the front and rear axles, $E$ is wheel track (average of front and rear track), $\delta_{fl}$ and $\delta_{fr}$ are the front left wheel angle and front right wheel angle, supposing $\delta_{fl} = \delta_{fr} = \delta$, $\beta$ is centroid slip angle, $V_g$ is the centroid velocity, $\dot{\psi}$ is yaw rate, $I_z$ is the moment of inertia of the vehicle about the *z*-axis, $V_x$ and $V_y$ are vehicle longitudinal and lateral speed, $F_{xi}$ and $F_{yi}$ are wheel longitudinal and lateral forces ($I = fl, fr, rl, rr$), $\alpha_i$ is tire slip angle ($I = fl, fr, rl, rr$), $F_{zi}$ is vehicle normal force ($I = fl, fr, rl, rr$), $f_i$ is rolling resistance, ($I = fl, fr, rl, rr$), $\rho$ is air quality factor, $A$ is the windward area of the vehicle, that is, the shadow area of the direction in which the vehicle is traveling, $C_D$ is air drag coefficient, $u_r$ is relative speed, that is, the speed of the vehicle when there is no wind.

### 2.2. Tire Normal Force Calculation

The normal force of the tire is mainly generated by the total mass of the vehicle. When the vehicle is running, longitudinal acceleration and deceleration change the normal force of the wheel. When the vehicle accelerates, the normal force on the front axle decreases and the normal force on the rear axle increases. When the vehicle turns, the normal forces on the left and right wheels of the vehicle change. The tire normal force can be calculated from the vehicle static model in Figure 3.

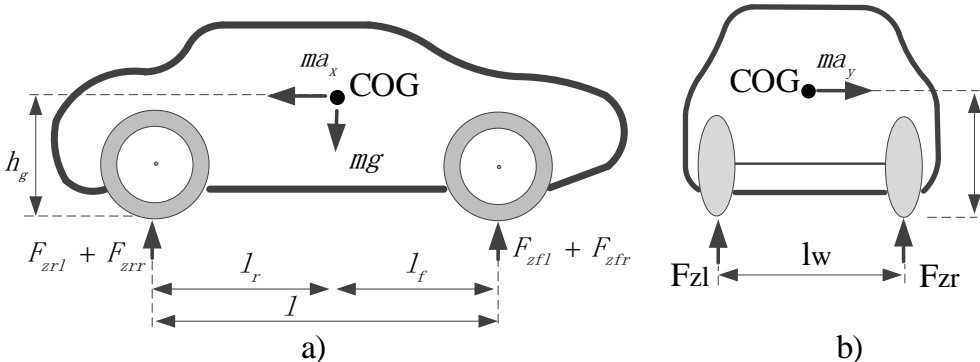

**Figure 3.** (**a**) Wheel load transfer at positive longitudinal acceleration and (**b**) front axle wheel load variation due to lateral acceleration during left turn.

As shown in Figure 3, the normal force of each wheel can be obtained from the static vehicle model:

$$F_{zfl} = \frac{m_s g l_r - m_s a_x h}{2L} - \frac{m_s a_y h l_r}{L l_w} + \frac{m_u g}{4} \tag{4}$$

$$F_{zfr} = \frac{m_s g l_r - m_s a_x h}{2L} + \frac{m_s a_y h l_r}{L l_w} + \frac{m_u g}{4} \tag{5}$$

$$F_{zrl} = \frac{m_s g l_f + m_s a_x h}{2L} - \frac{m_s a_y h l_f}{L l_w} + \frac{m_u g}{4} \tag{6}$$

$$F_{zrr} = \frac{m_s g l_f + m_s a_x h}{2L} + \frac{m_s a_y h l_f}{L l_w} + \frac{m_u g}{4} \tag{7}$$

where $m_s$ is sprung mass, $m_u$ is unsprung mass, $l_f$ is distance from mass center to front axle, $l_r$ is distance from the center of mass to rear axle, $g$ is gravitational acceleration, $a_x$ and $a_y$ are longitudinal and lateral acceleration, $h$ is the height of vehicle mass center, $l_w$ is the left and right wheelbase, $L$ is the front and rear vehicle wheelbase.

### 2.3. Wheel Slip Rate Calculation

Let the longitudinal speed of the tire axis be $V_w$, and the linear speed of the tire in the direction of rotation be $R_{eff}\omega_w$, where $R_{eff}$ is the effective radius of the tire, and $\omega_w$ is the angular velocity of the tire, as shown in Figure 4. The slip rate of a tire is defined as:

Longitudinal movement direction

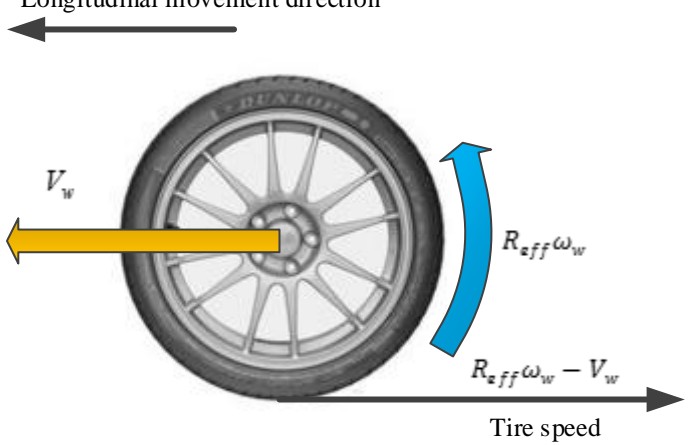

**Figure 4.** Tire longitudinal force diagram.

When the vehicle is accelerating:

$$s = \frac{R_{eff}\omega_w - V_w}{R_{eff}\omega_w} \tag{8}$$

When the vehicle is braking:

$$s = \frac{R_{eff}\omega_w - V_w}{V_w} \tag{9}$$

### 2.4. Tire Effective Model Construction

When the wheel is stationary, the vertical load applied by the body is applied, and its static radius $R_{stat}$ is:

$$R_{stat} = R_0 - \frac{F_z}{k_t} \tag{10}$$

In the equation, $R_0$ is the radius of the tire that is not deformed, $F_z$ is the vertical load on the tire and the ground contact surface at the station, $k_t$ is the tire vertical stiffness.

The relationship between the tire effective radius $R_{eff}$, the tire rotation angle velocity $\omega$ and tire speed is as follows:

$$V_{\omega x} = R_{eff}\omega \tag{11}$$

As shown in Figure 5, the length of the tire and the ground contact surface is $2a$, the wheel, the ground contact surface to the tire center and the wheel, and the end contact surface end to the center of the tire is $\phi$. Tire movement a distance takes $t$.

$$V_{\omega x} = R_{eff}\omega = \frac{a}{t} \tag{12}$$

At the same time, the rotational speed of the wheel is:

$$\omega = \frac{\phi}{t} \tag{13}$$

Calculating the Equations (12) and (13):

$$R_{eff} = \frac{a}{\phi} \tag{14}$$

From Figure 5:

$$R_{stat} = R_0 \cos\phi \tag{15}$$

$$a = R_0 \sin\phi \tag{16}$$

The effective radius of tires can be obtained from Equations (14)–(16):

$$R_{eff} = \frac{R_0 \sin\left[\arccos\left(\frac{R_{stat}}{R_0}\right)\right]}{\arccos\left(\frac{R_{stat}}{R_0}\right)} \tag{17}$$

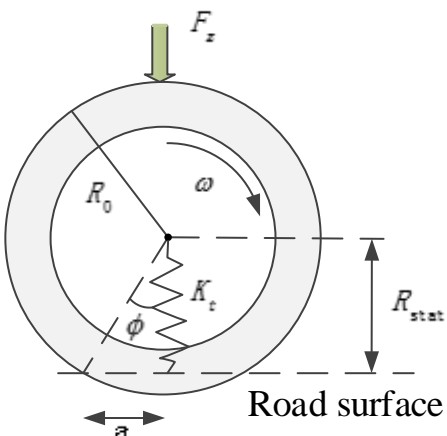

**Figure 5.** Static and dynamic wheel radius.

*2.5. Longitudinal Speed at Wheel Axle Calculation*

It can be seen from Figure 5 that, according to the vertical, lateral vehicle speed, the horizontal angle velocity and the wheel corner, the longitudinal velocity component of each wheel axis can be obtained from the following equations:

$$V_{wfl} = \left( V_x - \frac{\dot{\psi}E}{2} \right) \cos\delta + \left( V_y + \dot{\psi}l_f \right) \sin\delta \tag{18}$$

$$V_{wfr} = \left( V_x + \frac{\dot{\psi}E}{2} \right) \cos\delta + \left( V_y + \dot{\psi}l_f \right) \sin\delta \tag{19}$$

$$V_{wrl} = V_x - \frac{\dot{\psi}E}{2} \tag{20}$$

$$V_{wrr} = V_x + \frac{\dot{\psi}E}{2} \tag{21}$$

## 3. The Tire/Road Friction Estimation Algorithm

*3.1. Recursive Least Squares with Fixed Forgetting Factor*

The values obtained by using the normal force, the longitudinal force of the wheel, the speed at the center of the wheel and the wheel slip rate in Section 2 are the inputs of the RLS method with a fixed forgetting factor proposed in this section, and the output is the friction coefficient.

The relationship between the longitudinal force $F_{xf}$ of the vehicle front axle, the normal force $F_{zf}$ of the vehicle front axle and the mean value $s$ of the tire slip rate of the vehicle front axle is:

$$\frac{F_{xf}}{F_{zf}} = Ks \tag{22}$$

Basic form of RLS with a fixed forgetting factor can be expressed as:

$$y(n) = \varphi^T(n)\theta(n) + e(n) \tag{23}$$

In the formula, $y(n) = \frac{F_{xf}}{F_{zf}}$ is the system output; $\theta(n) = K$ is the unknown quantity; $\varphi(n) = s$ is the system input; $e(n)$ is the deviation.

The RLS method with stable forgetting factor is as follows:

$$e(n) = y(n) - \overset{\wedge}{\theta}^T (n-1)\varphi(n) \tag{24}$$

$$K(n) = \frac{P(n-1)\varphi(n)}{\lambda + \varphi^T(n)P(n-1)\varphi(n)} \tag{25}$$

$$\hat{\theta}(n) = \hat{\theta}(n-1) + K(n)e(n) \tag{26}$$

$$P(n) = \frac{1}{\lambda}\left[I - K(n)\varphi^T(n)\right]P(n-1) \tag{27}$$

In the equations, $e(n)$ is the deviation signal, $y(n)$ is the system output, $\varphi^T(n)$ is the observation matrix and $\hat{\theta}(n-1)$ is the estimation value received by recursion $n-1$ times. $K(n)$ is the Kalman filter gain matrix, $P(n)$ is the covariance matrix and its initial value is $P(0) = 10^\beta I$, where $I$ is the identity matrix, $\beta$ generally takes a larger positive integer, $\lambda$ is the forgetting factor, and the value is between 0 and 1.

*3.2. Estimation of Pavement Friction Coefficient Based on VFF-RLS*

In the traditional RLS algorithm, $\lambda$ is a stable value, that is, the effect of past observation data is weakened at a stable rate. If $\lambda$ is close to 1, RLS has high accuracy but the ability to track parameters is reduced. Decreasing $\lambda$ can improve the tracking ability, but at the same time, reduce the steady-state accuracy. In order to meet these two contradictory needs, a friction coefficient estimation algorithm based on VFF-RLS is proposed.

The $e(n)$ in Equation (24) is calculated according to the estimation value of the $n-1$th cycle, which is a priori deviation. The posterior deviation can be defined as:

$$\varepsilon(n) = y(n) - \hat{\theta}^T(n)\varphi(n) \tag{28}$$

From Equations (24), (26) and (28) we can get:

$$\varepsilon(n) = e(n)\left[1 - \varphi^T(n)K(n)\right] \tag{29}$$

The forgetting factor is established by restoring the system noise in the deviation signal, that is, the forgetting factor $\lambda(n)$ can be calculated according to Equation (30):

$$E\left\{\varepsilon^2(n)\right\} = E\left\{v^2(n)\right\} \tag{30}$$

where $E\left\{v^2(n)\right\} = \sigma_v^2$ is the power of system noise.

Substitute Equations (25) and (29) into Equation (30) to get:

$$E\left\{\left[\frac{\lambda(n)}{\lambda(n) + q(n)}\right]^2\right\} = \frac{\sigma_v^2}{\sigma_e^2(n)} \tag{31}$$

where $q(n) = \varphi^T(n)P(n-1)\varphi(n)$ and $E\left\{e^2(n)\right\} = \sigma_e^2(n)$ are the powers of the prior deviation signal. In Equation (31), we assume that the input signal and the deviation signal are unrelated. This assumption is valid when the identification parameters begin to converge to the actual values. By solving Equation (31), the expression for the variable forgetting factor can be obtained:

$$\lambda(n) = \frac{\sigma_v\sigma_q(n)}{\sigma(n) - \sigma_v} \tag{32}$$

where the power estimate $E\left\{q^2(n)\right\} = \sigma_q^2(n)$ can be obtained from Equations (33) and (34):

$$\hat{\sigma}_e^2(n) = \alpha\hat{\sigma}_e^2(n-1) - (1-\alpha)e^2(n) \tag{33}$$

$$\hat{\sigma}_q^2(n) = \alpha\hat{\sigma}_q^2(n-1) - (1-\alpha)q^2(n) \tag{34}$$

where $\alpha$ is the weighting factor.

Considering that the value of $\lambda$ must be in the range of [0, 1], the forgetting factor proposed in this paper can be given as follows:

$$\lambda(n) = \min\left\{ \frac{\sigma_v \hat{\sigma}_q(n)}{\xi + \left| \hat{\sigma}_e(n) - \sigma_v \right|}, \lambda_{\max} \right\} \tag{35}$$

where $\xi$ is a small positive number that prevents division by zero. Before the algorithm converges or when there is a sudden change in the system, $\hat{\sigma}_e(n)$ is larger than $\sigma_v$, so that $\lambda(n)$ is a lower value, so that the algorithm can enable fast convergence. When the algorithm converges to a steady state solution, $\lambda(n)$ is transformed into $\lambda_{\max}$, which makes the algorithm have higher estimation accuracy.

### 3.3. Estimation Model of Road Friction Coefficient

The modeling process mainly includes the calculation of the wheel normal force, wheel longitudinal force calculation, speed calculation at the wheel center, wheel slip rate calculation and estimation of the friction coefficient by VFF-RLS. The basic framework of Matlab/Simulink modeling is shown in Figure 6.

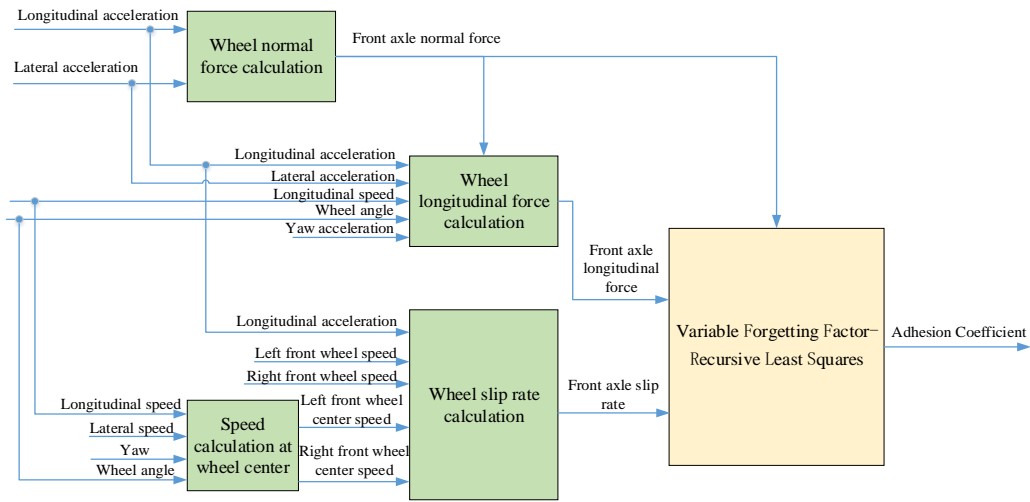

**Figure 6.** Road friction coefficient calculation process.

The overall model of the friction coefficient estimation algorithm is shown in Figure 7. The input includes: wheel normal force, wheel longitudinal force and wheel slip rate. Finally, the RLS with forgetting factor is used to estimate the friction coefficient. The module 1 in the figure is the wheel normal force calculation module, the wheel slip rate calculation module and the wheel longitudinal force calculation module, and the module 2 in the figure is the road friction coefficient estimation module.

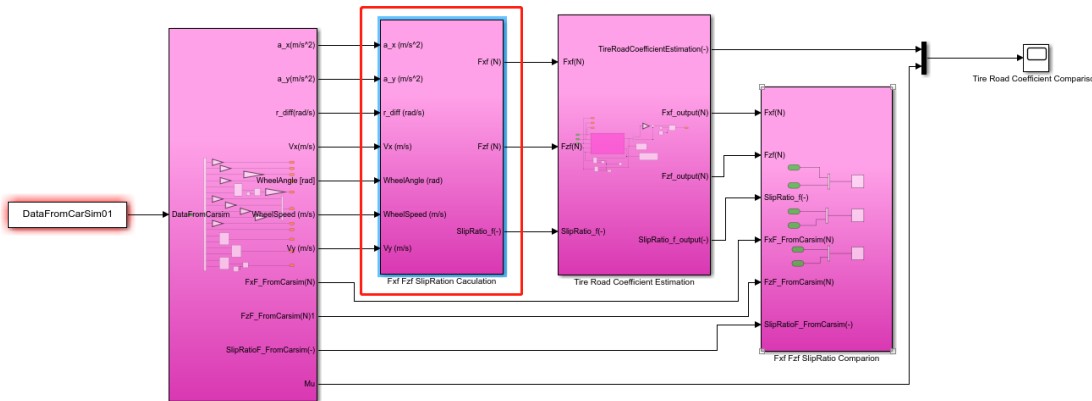

**Figure 7.** Overall model of pavement friction coefficient estimation algorithm in Simulink environment.

## 4. Trajectory Tracking Control Based on MPC

The design of the MPC-based tracking control system is the core control algorithm of the entire controller. The structure of the algorithm can be divided into the following three parts: (1) Rewrite the prediction model, namely the lateral dynamics model, into discrete state space equations; (2) Construct the objective function of the trajectory tracking algorithm and the constraints of each control quantity; (3) Convert each control objective into a standard quadratic programming form to solve.

### 4.1. Establishment of Vehicle Model

The traditional bicycle model formula is rewritten into the form of the state space equation shown in Equation (36) below, where the state quantity is $x = \left[y, a_y, \psi, \dot{\psi}\right]$, the control input is $u = \delta_f$,

$$\dot{x}(k) = Ax(k) + Bu(k) \tag{36}$$

Among them, $y$ is the lateral displacement, $a_y$ is the lateral velocity, $\psi$ is the yaw angle, $\dot{\psi}$ is the yaw rate and the control input $\delta_f$ is the tire rotation angle.

$$A = \begin{bmatrix} 0 & 1 & \dot{x} & 0 \\ 0 & -2\frac{k_{\alpha f}c_{\alpha f}+k_{\alpha r}c_{\alpha r}}{m\dot{x}} & 0 & 2\frac{l_r k_{\alpha f}c_{\alpha r}-l_f k_{\alpha f}c_{\alpha f}}{m\dot{x}} - \dot{x} \\ 0 & 0 & 0 & 1 \\ 0 & 2\frac{l_r k_{\alpha r}c_{\alpha r}-l_f k_{\alpha f}c_{\alpha f}}{I_Z \dot{x}} & 0 & -2\frac{l_f^2 k_{\alpha f}c_{\alpha f}-l_r^2 k_{\alpha r}c_{\alpha r}}{I_Z \dot{x}} \end{bmatrix}, \ B = \begin{bmatrix} 0 \\ 2\frac{k_{\alpha f}c_{\alpha f}}{m} \\ 0 \\ 2\frac{l_f k_{\alpha r}c_{\alpha r}}{I_Z} \end{bmatrix} \tag{37}$$

where $k_{\alpha f}$ is the front-wheel cornering stiffness adjustment coefficient, $k_{\alpha r}$ the rear-wheel cornering stiffness adjustment coefficient, $c_{\alpha f}$ is the front-wheel cornering stiffness, $c_{\alpha r}$ is the rear-wheel cornering stiffness.

Since the saturated section of the tire characteristic curve cannot be expressed by the linear tire model formula, the linear tire model needs to be improved. In this paper, a tire cornering stiffness adjustment coefficient $k_\alpha$ is introduced, so the improved tire model expression is:

$$F_{yf} = k_{\alpha f}c_{\alpha f}\alpha_f$$
$$F_{yr} = k_{\alpha r}c_{\alpha r}\alpha_r$$

Equation (37) is approximately discretized, and the processing form is shown in Equation (38):

$$A = I + TA$$
$$B = TB \tag{38}$$

The new discretized state space equation is received as follows:

$$x(k+1) = Ax(k) + Bu(k) \tag{39}$$

The discretized state space equation of the bicycle model is in the form of Equation (38), where:

$$A = \begin{bmatrix} 0 & T_s & \dot{x}T_s & 0 \\ 0 & -2\frac{k_{\alpha f}c_{\alpha f}+k_{\alpha r}c_{\alpha r}}{m\dot{x}}T_s & 0 & 2\frac{l_r k_{\alpha r}c_{\alpha r}-ak_{\alpha f}c_{\alpha f}}{m\dot{x}}T_s - \dot{x}T_s \\ 0 & 0 & 0 & 1 \\ 0 & 2\frac{l_r k_{\alpha r}c_{\alpha r}-ak_{\alpha f}c_{\alpha f}}{I_z \dot{x}}T_s & 0 & -2\frac{l_f^2 k_{\alpha f}c_{\alpha f}+l_r^2 k_{\alpha r}c_{\alpha r}}{I_z \dot{x}}T_s + 1 \end{bmatrix} \tag{40}$$

*4.2. Establishment of Vehicle Model*

The core goal of the trajectory tracking system is to eliminate the lateral offset deviation of the vehicle to ensure that the vehicle follows the referenced trajectory. At the same time, it is also necessary to ensure the stability of the vehicle during driving and avoid the occurrence of lane departure due to excessive rotation angle. Therefore, it is also necessary to limit the control input, that is, the increment of the front wheel rotation angle, to ensure the smoothness of the control process. So, the control objective is shown in Equation (41):

$$Objective : \left(y_{ref} - \widetilde{y}\right) \to 0, \Delta u \to 0 \tag{41}$$

where $y_{ref}$ is the desired trajectory, $\widetilde{y}$ is the prediction output in the prediction time domain; $\Delta u$ is the front wheel rotation angle increment.

Therefore, the objective function of the trajectory tracking controller design based on MPC is:

$$J(k) = \left(Y_{ref} - \widetilde{y}(k)\right)^T Q\left(Y_{ref} - \widetilde{y}(k)\right) + \Delta \widetilde{u}^T(k)R\Delta \widetilde{u}(k) \tag{42}$$

Among them, $Q$ and $R$ represent the weight matrix. The first term reflects the performance of target tracking, that is, making the predicted output as close to the expected output as possible, while the second term limits the variation of the control increment.

The advantage of the MPC controller is that it can directly handle various constraints. When the vehicle travels under certain extreme conditions, such as under wet and slippery roads, if the vehicle speed is too high, there will be dangers such as body side slippage. At this time, in addition to achieving zero lateral offset deviation and the proper front wheel turning angle, the stability and safety of vehicle driving should also be ensured. At this time, it is necessary to constrain the state of the vehicle, such as the tire slip angle constraints.

Considering the limitation of the actual steering structure of the vehicle, the front wheel turning angle needs to be constrained, as shown in Equation (43):

$$\delta_{\min} \leq \delta(k) \leq \delta_{\max} \tag{43}$$

Considering the stability and comfort of the driving process, it is necessary to constrain the front wheel angle increment to avoid dangerous situations such as sharp turns in the control process, so the angle increment constraints need to be imposed as follows:

$$\Delta\delta_{\min} \leq \Delta\delta(k) \leq \Delta\delta_{\max} \tag{44}$$

When the vehicle is driven under certain extreme conditions, such as driving on a slippery road at a higher speed, the vehicle will become unstable, because the grip of the tire is at the maximum at this time, so it cannot provide a large enough force. Therefore, in this paper, the tire slip angle is properly constrained in the control process, and the expression is as follows:

$$\alpha_{\min} \leq \alpha_{f,r} \leq \alpha_{\max} \tag{45}$$

where $\alpha_{\min}$ and $\alpha_{\max}$ are the lower and upper bounds of the tire slip angle, respectively.

### 4.3. MPC Controller Solver

Since the control target of the trajectory tracking system is the front-wheel steering angle increment $\Delta u$:

$$\begin{bmatrix} x(k+1) \\ u(k) \end{bmatrix} = \begin{bmatrix} A & B \\ 0 & I \end{bmatrix} \begin{bmatrix} x(k) \\ u(k-1) \end{bmatrix} + \begin{bmatrix} B \\ I \end{bmatrix} \Delta u(k) \tag{46}$$

$$y(k) = \begin{bmatrix} C & 0 \end{bmatrix} \begin{bmatrix} x(k) \\ u(k-1) \end{bmatrix} \tag{47}$$

among them, C = [1,0,0,0], $y(k)$ is the lateral displacement in the body coordinate system, which is:

$$x_m(k+1) = A_m x_m(k) + B_m \Delta u(k) \tag{48}$$

$$y(k) = C_m x_m(k) \tag{49}$$

among them, $A_m = \begin{bmatrix} A & B \\ 0_{m*n} & I_m \end{bmatrix}$, $B_m = \begin{bmatrix} B \\ I_m \end{bmatrix}$, $n$ is the state dimension, $m$ is the control dimension.

The prediction output of the prediction time domain at each moment can be deduced as follows:

$$\begin{aligned}
y(k) &= C_m x_m(k) \\
y(k+1) &= C_m x_m(k+1) = C_m A_m x_m(k) + C_m B_m \Delta u(k) \\
y(k+2) &= C_m x_m(k+2) = C_m A_m^2 x_m(k) + C_m A_m B_m \Delta u(k) + C_m B_m \Delta u(k+1) \\
&\vdots \\
y(k+N_p) &= C_m x_m(k+N_p) = C_m A_m^{N_p} x_m(k) + C_m A_m^{N_p-1} B_m \Delta u(k) \\
&+ C_m A_m^{N_p-2} B_m \Delta u(k+1) + \cdots + C_m A_m^{N_p-N_c} B_m \Delta u(k+N_c-1)
\end{aligned}$$

In the above equation, $N_p$ is the prediction time domain in model predictive control, and $N_c$ is the control time domain. From the above process, it can be concluded that the predicted output and control quantity of the system in the prediction time domain are respectively:

$$\widetilde{y}(k) = \begin{bmatrix} y(k+1) & y(k+2) & \cdots & y(k+m) & \cdots & y(k+N_p) \end{bmatrix}^T \tag{50}$$

$$\Delta \widetilde{u}(k) = \begin{bmatrix} \Delta u(k) & \Delta u(k+1) & \cdots & \Delta u(k+N_c-1) \end{bmatrix}^T \tag{51}$$

Therefore, the predicted output in the control time domain of the system can be rewritten as follows:

$$\widetilde{y}(k) = F x_m(k) + \Phi \Delta \widetilde{u}(k) \tag{52}$$

where,

$$F = \begin{bmatrix} C_m A_m & C_m A_m^2 & C_m A_m^3 & \cdots & C_m A_m^{N_p} \end{bmatrix}^T,$$

$$\Phi = \begin{bmatrix}
C_m B_m & 0 & 0 & \cdots & 0 \\
C_m A_m B_m & C_m B_m & 0 & \cdots & 0 \\
C_m A_m^2 B_m & C_m A_m B_m & C_m B_m & \cdots & 0 \\
\vdots & \vdots & \vdots & \ddots & \vdots \\
C_m A_m^{N_p-1} B_m & C_m A_m^{N_p-2} B_m & C_m A_m^{N_p-3} B_m & \cdots & C_m A_m^{N_p-N_c} B_m
\end{bmatrix}$$

Since the objective function Equation (42) is a general form, in order to facilitate the computer's solution operation, it needs to be properly processed into a standard quadratic

form, that is, quadratic programming (Quadratic Programming, QP) problem. The standard format of quadratic programming is as follows:

$$\min \frac{1}{2} \Delta \widetilde{u}^T(k) H \Delta \widetilde{u}(k) + b \Delta \widetilde{u}(k) \tag{53}$$

Take Equation (52) into Equation (53) and rewrite it in the quadratic programming standard format, where $H = \Phi^T Q \Phi + R$, $b = 2 \left( F x_m(k) - Y_{ref} \right)^T Q \Phi$.

For constraints in the control process, constraints (44) can be rewritten in the form of inequalities related to $\Delta \widetilde{u}$, that is $\begin{bmatrix} -I_m \\ I_m \end{bmatrix} \Delta \widetilde{u} \leq \begin{bmatrix} -\Delta \widetilde{u}_{min} \\ \Delta \widetilde{u}_{max} \end{bmatrix}$. Similarly, constraints (43) can be rewritten in the form of inequalities related to $\Delta \widetilde{u}$, that is:

$$\begin{bmatrix} -C_2 \\ C_2 \end{bmatrix} \Delta \widetilde{u} \leq \begin{bmatrix} -\widetilde{u}_{min} + C_1 u(k-1) \\ \widetilde{u}_{max} - C_1 u(k-1) \end{bmatrix} \tag{54}$$

where,

$$C_1 = \begin{bmatrix} I_m \\ I_m \\ I_m \\ \vdots \\ I_m \end{bmatrix}_{mN_c \times m}, \quad C_2 = \begin{bmatrix} I_m & 0_m & 0_m & \cdots & 0_m \\ I_m & I_m & 0_m & \cdots & 0_m \\ I_m & I_m & I_m & \cdots & 0_m \\ \vdots & \vdots & \vdots & \ddots & \vdots \\ I_m & I_m & I_m & \cdots & I_m \end{bmatrix}_{mN_c \times mN_c}$$

The relationship between the tire slip angle and the vehicle state $x$ and the control input $\Delta u$ can be expressed as:

$$\alpha_f = \delta_f - \frac{\dot{y} + l_f \dot{\varphi}}{\dot{x}} = C_f x + D_f U = C_f x + D_f (C_1 u_{k-1} + C_2 \Delta u_k) \tag{55}$$

$$\alpha_r = -\frac{\dot{y} - l_r \dot{\varphi}}{\dot{x}} = C_r x \tag{56}$$

where,

$$C_f = \frac{1}{-v_x} \begin{bmatrix} 0 & 1 & 0 & l_f \end{bmatrix}, C_r = \frac{1}{v_x} \begin{bmatrix} 0 & -1 & 0 & l_r \end{bmatrix}, D_f = \begin{bmatrix} 0 & 1 \end{bmatrix}$$

If the state quantity $x$ at time $k$ and the control quantity $\Delta u_{k-1}$ at the previous time are known, and the optimal solution is obtained from Equation (53) in the control time domain, the optimal control increment sequence in the control time domain can be obtained as $\Delta \widetilde{u}^*(k) = \begin{bmatrix} \Delta u^*(k) & \Delta u^*(k+1) & \cdots & \Delta u^*(k+N_c-1) \end{bmatrix}^T$. From the above-mentioned principle of model predictive control, it can be known that the optimal control sequence will be obtained in each control process, and then the first item in this sequence will act on the system as the actual control increment, that is:

$$u(k) = u(k-1) + \begin{bmatrix} I_m & 0_m & \cdots & 0_m \end{bmatrix}^T \Delta \widetilde{u}^*(k) \tag{57}$$

As shown in Equation (57), it is the control input of the system at the current moment, that is, the front-wheel steering angle.

## 5. Simulation Results

### 5.1. Simulation Verification of Pavement Friction Coefficient Estimation Model

The slip ratio of the wheel can be divided into two cases: high slip ratio (slip ratio greater than 0.01) and low slip ratio. Firstly, the simulation under the condition of the high slip rate is carried out, and the single condition of straight driving and uniform acceleration is adopted in the simulation. When the vehicle acceleration is 1.2 m/s$^2$, and the driving condition is straight-line driving, the road friction coefficient is estimated from 15 s.

The slip ratio of the vehicle front axle is shown in Figure 8. When the road friction coefficient is 0.3–1, the slip rate calculated in this paper is basically consistent with the slip rate output by Carsim. When the road friction coefficient is 0.1 and 0.2, there is a big difference between the slip ratio calculated according to Equations (8) and (9) and the slip ratio output by Carsim. The reason is that the calculation equations of the two are different. The Carsim simulation software adopts the equation $s = \frac{\omega R - V_x}{|V_x|}$ (where $\omega$ is the wheel speed, $R$ is the effective radius of the tire and $V_x$ is the wheel center speed).

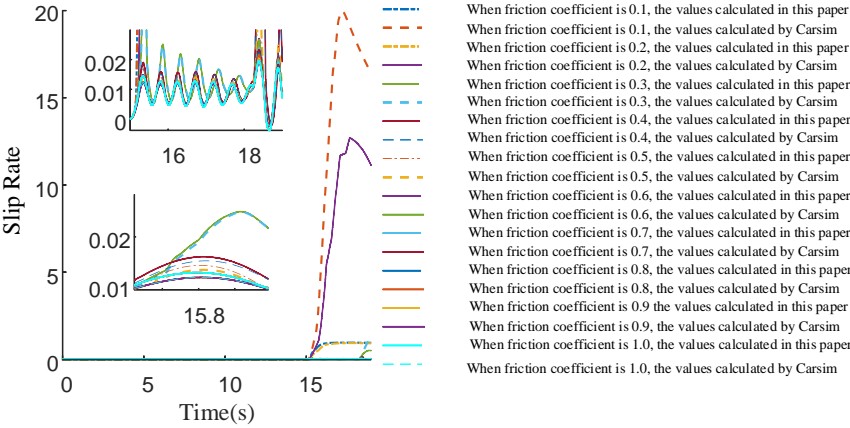

**Figure 8.** Tire front axle slip rate (average of two tires).

As shown in Figure 9, in Carsim, the simulation conditions in this section select two test roads with a friction coefficient of 0.8 and 0.1, respectively. The other test conditions are the same as the previous ones. It is concluded that the road friction coefficient estimator based on VFF-RLS in this paper can better track the friction coefficient value set in the Carsim software. However, the friction coefficient estimator based on the RLS algorithm will have a certain degree of overshoot and lag. Through the curve comparison, it can be found that the estimation effect of the road friction coefficient estimator based on the VFF-RLS proposed in this paper is better than the estimator based on the RLS.

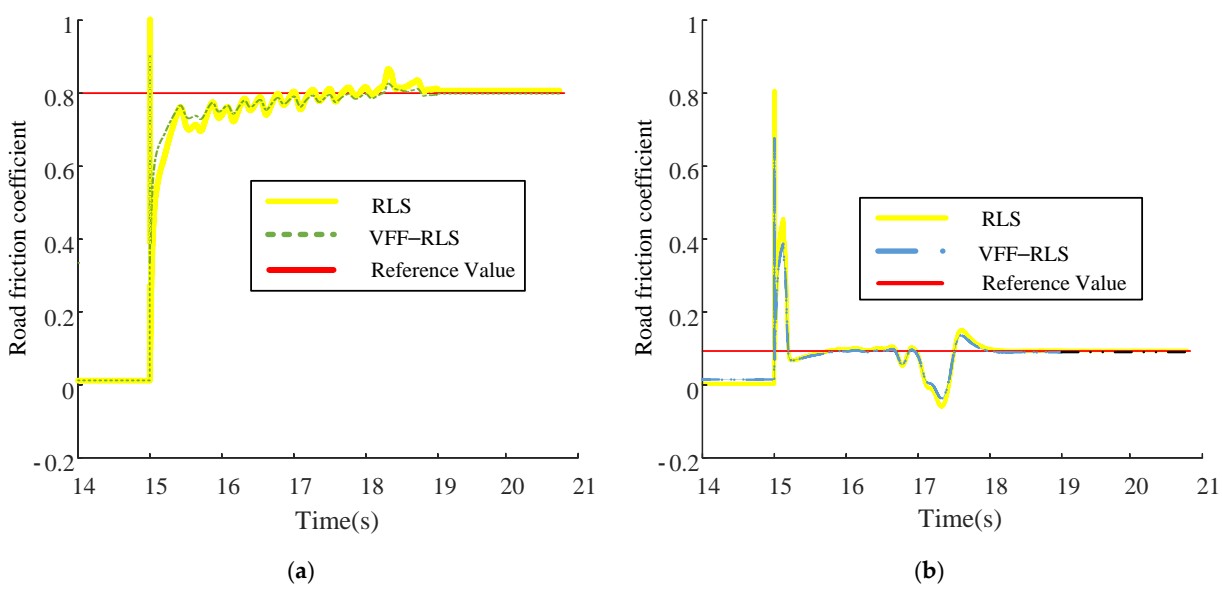

(**a**)  (**b**)

**Figure 9.** Estimation of road friction coefficient. (**a**) When road friction coefficient is 0.8 (**b**) When road friction coefficient is 0.1.

### 5.2. Trajectory Tracing Controller Simulation Verification

In the co-simulation environment of CarSim and MATLAB/Simulink, the above-mentioned road adhesion coefficient estimation strategy and trajectory tracking control

strategy are constructed. The vehicle model parameters used in the CarSim vehicle dynamics simulation are shown in Table 1.

**Table 1.** Vehicle parameters.

| Symbol | Dimension | Value |
|---|---|---|
| $m$ | kg | 1296 |
| $m_s$ | kg | 1200 |
| $m_u$ | kg | 96 |
| $I_z$ | kg·m$^2$ | 1750 |
| $h_g$ | m | 0.54 |
| $l_f$ | m | 1.25 |
| $l_r$ | m | 1.32 |
| $l_w$ | m | 1.405 |
| $R_0$ | m | 0.315 |
| $k_t$ | kN/m | 100 |
| $c_{\alpha f}$ | N/rad | 66,900 |
| $c_{\alpha r}$ | N/rad | 62,700 |

In this paper, two different simulation test conditions are established to test and verify the validity of the road friction coefficient estimation strategy and the trajectory tracking controller constructed above. In order to simplify the test conditions, the running speed is designed as a fixed value. Scenario A describes the performance of vehicles turning right at the intersection on the road friction coefficient of 0.8 at the speed of 60 km/h, and Scenario B presents of vehicles turning right at the intersection on the road friction coefficient of 0.1 at the speed of 60 km/h.

(1)  Scenario A

As shown in Figures 10–13, when the vehicle is in Scenario A, whether it is the steering wheel angle, the driving trajectory or the yaw rate, it can better track the referenced value of the intelligent vehicle. Among them, the lateral trajectory maximum deviation is (−1.0 m, 0.5 m) range. Thus, the outstanding trajectory tracking performance of the intelligent vehicle is achieved. However, large control overshoot and control time lag appear in the RLS-based MPC control system. In contrast, the control method proposed in this paper has better performance in accuracy and real-time. Therefore, the MPC controller based on VFF-RLS has better control effect.

(2)  Scenario B

As shown in Figures 14–17, when the vehicle is in Scenario B, whether it is the steering wheel angle, the driving trajectory or the yaw rate, it can better track the referenced trajectory. Among them, the lateral trajectory maximum deviation is (−1.0 m, 0.8 m) range. As for lateral deviation, the control result under MPC control with the VFF-RLS method is obviously smaller than that of the MPC control with the RLS method, and the deviation is controlled within a small range and is relatively stable. Thus, the outstanding trajectory tracking performance of the intelligent vehicle is achieved. In addition, large control overshoot and control time lag appear in the RLS-based MPC control system. Therefore, the MPC controller based on VFF-RLS has better control effect.

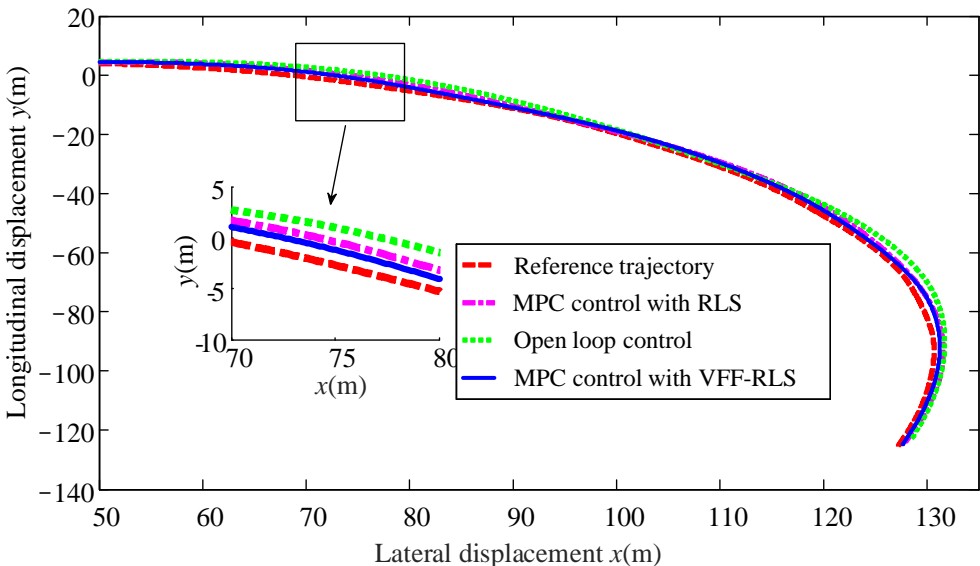

**Figure 10.** Vehicle trajectory under Scenario A.

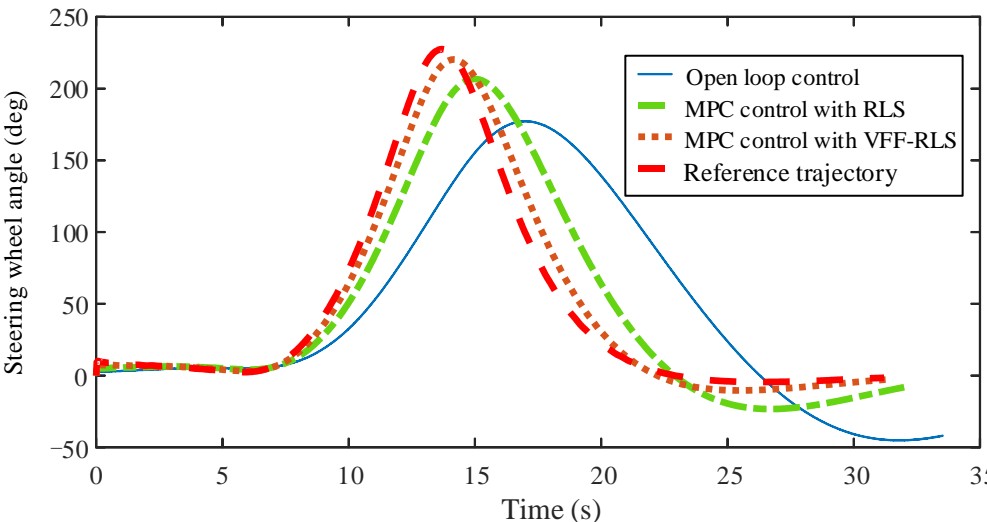

**Figure 11.** Steering wheel angle under Scenario A.

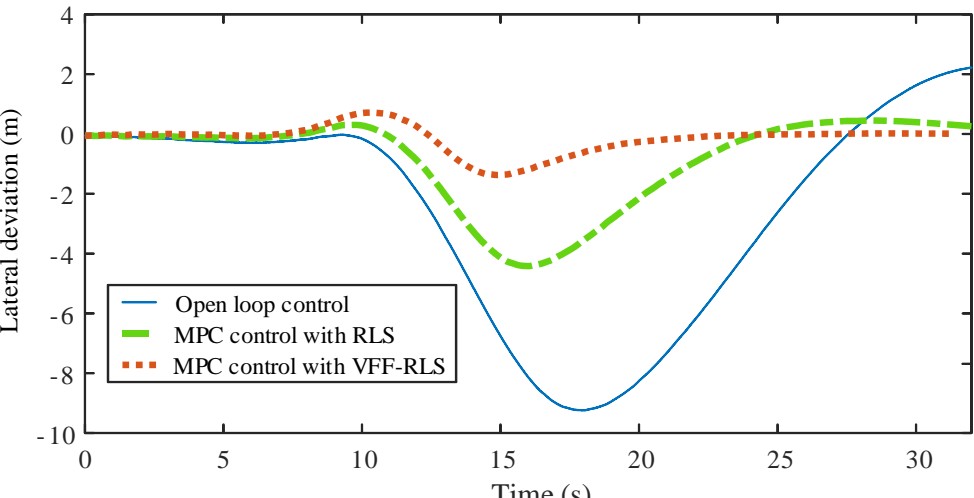

**Figure 12.** Lateral deviation under Scenario A.

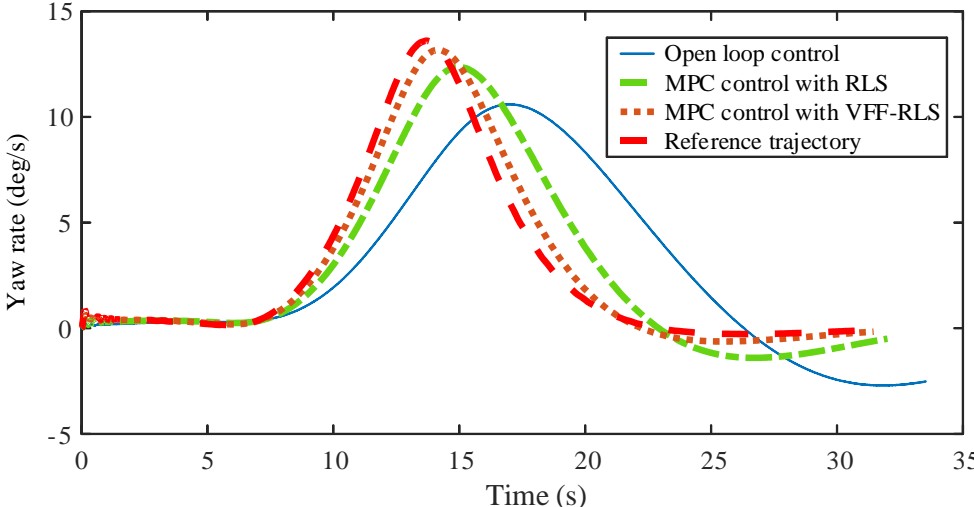

**Figure 13.** Yaw rate under Scenario A.

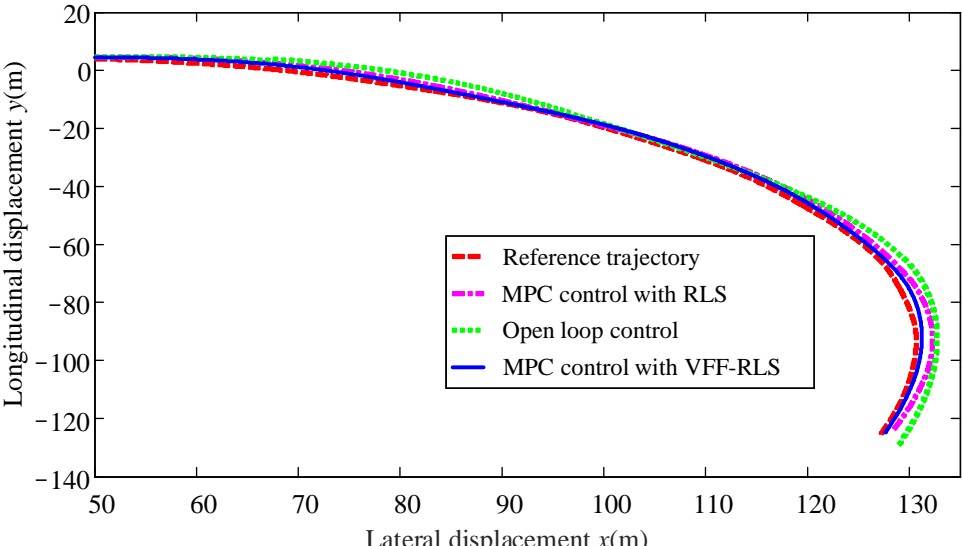

**Figure 14.** Vehicle trajectory under Scenario B.

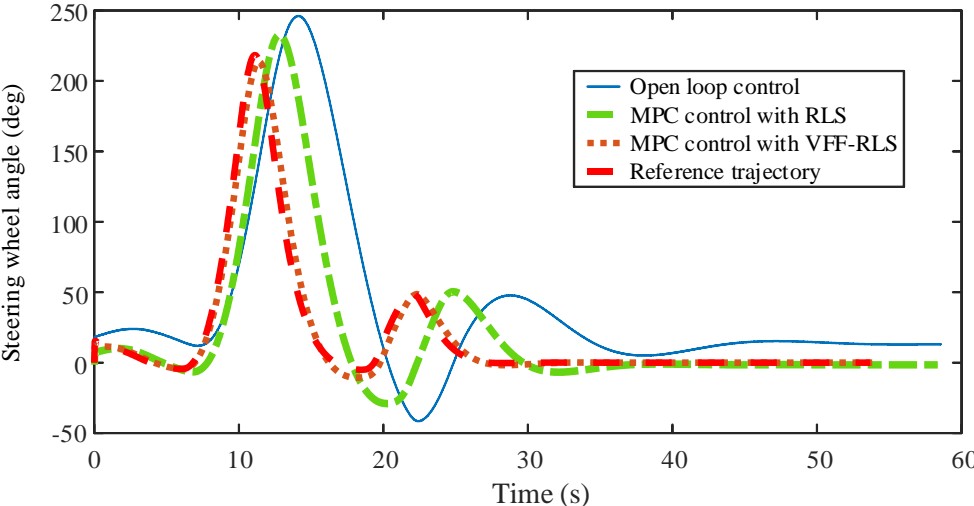

**Figure 15.** Steering wheel angle under Scenario B.

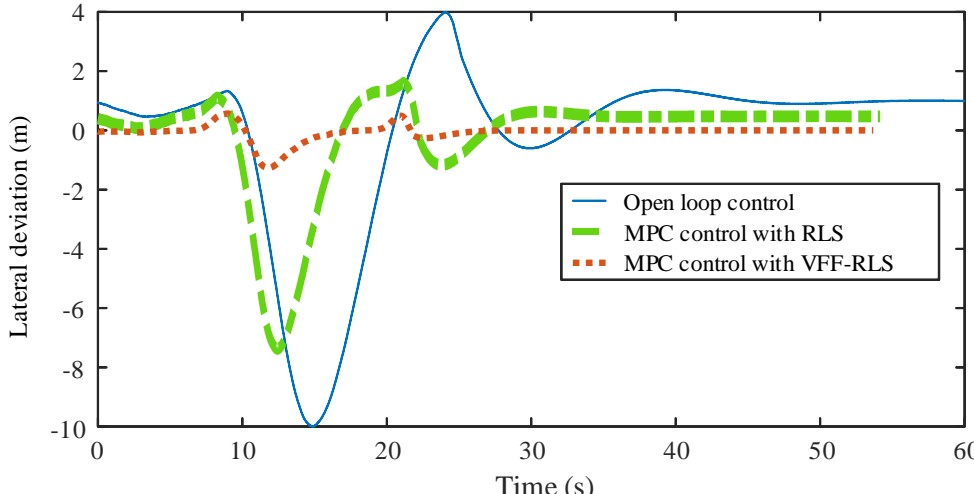

**Figure 16.** Lateral deviation under Scenario B.

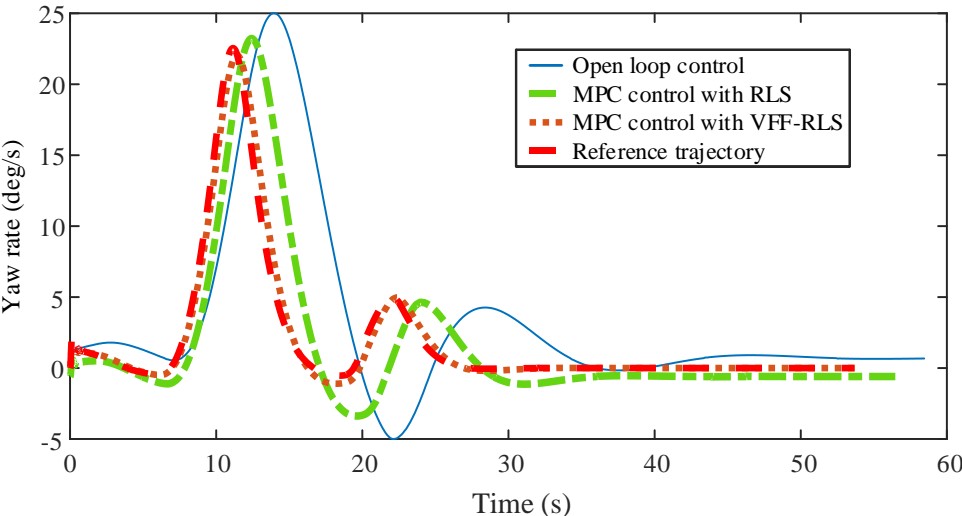

**Figure 17.** Yaw rate under Scenario B.

## 6. Conclusions

In this paper, a trajectory tracking control method based on VFF-RLS road friction coefficient estimation of an intelligent vehicle is presented for better control performance. The vehicle dynamics model and the tire equivalent model are constructed, then, this paper designs an estimation strategy of the road friction coefficient based on VFF-RLS, at the same time, on the basis of the estimation method, the identified road adhesion results are output to the model predictive controller constructed in this paper as a way to improve tire slip angle constraints to realize the trajectory tracking. The joint simulation test results show that the trajectory tracking control system based on VFF-RLS-based MPC proposed in this paper has outstanding tracking performance, which can realize vehicle stability control while ensuring vehicle trajectory tracking accuracy.

Future work can consider trajectory-tracking for an autonomous lane-changing control system. Lane-changing intention recognition in a complex traffic environment is another important direction.

**Author Contributions:** M.Z. made important contributions to the thinking and design of the research. Y.N., Y.H. and X.Z. have made important contributions to literature retrieval, chart making, data analysis and manuscript writing. All authors have read and agreed to the published version of the manuscript.

**Funding:** This work is supported by Tianjin Artificial Intelligence Innovation Fund (No. 17YDLJGX00020).

**Institutional Review Board Statement:** Not applicable.

**Informed Consent Statement:** Not applicable.

**Data Availability Statement:** The raw/processed is temporarily unavailable.

**Conflicts of Interest:** The authors declare no conflict of interest.

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
