# Peer review of "Intelligent Vehicle Trajectory Tracking Control Based on VFF-RLS Road Friction Coefficient Estimation"

_electronics, doi:10.3390/electronics11193119_

Round 1

Reviewer 1 Report

The introductory section is too long and has no real use for the reader who should be able to understand the objectives of this work and the concrete contributions of the authors with respect to the state of the art.

I strongly recommend a division into subsections, as given below.

Rationale -> in which the authors explicitly explain what kind of issue they want to address by explaining to the reader the importance of the issue and what the shortcomings of the state of the art are.

State of the art and related work-> in which the authors introduce the most recent work in the literature with which they critically compare with the work presented. Explicitly listing advantages and disadvantages.

Contributions-> in which the authors explicitly list the claims of this work and the actual contributions compared to the state of the art.

I note that many of the articles cited are pre-2018. For an applied research article such as this one, it is not acceptable for citations to predate the last 4 years of the state of the art. This is not a state-of-the-art review article, survey paper, or book chapter. I would beg the authors to include more recent articles such as the following:

https://ieeexplore.ieee.org/document/9805740

https://ieeexplore.ieee.org/document/9091169

https://link.springer.com/chapter/10.1007/978-3-030-66729-0_26

Throughout the text, mathematical expressions are written in a very haphazard manner. Please review this aspect. You should respect the font and font size of the MDPI template.

Many of the figures/schemas have illegible inner lettering. Please fix this aspect.

Section 3 introduces the estimation method via RLS "modified" in an overly operational way. Being the "centrepiece" of this paper, I expect a much more detailed discussion of formal aspects, such as convergence conditions and robustness to model uncertainties.

The current version reports only expressions (rather classical actually) without any real discussion of advantages and disadvantages compared to other model-based estimation methods (such as KF, EKF, UKF, etc.).

The description of the MPC control technique is too didactic.

I recommend taking cues from the works recommended above.

There is a lack of discussion on the choice of the main parameters of MPC such as prediction horizon, the proportion of weights, and numerical integration time within the algorithm. It is also unclear whether the authors use MPC or AMPC since speed explicitly appears in Eq.40 in the matrix.

Also completely missing is a discussion of the chosen numerical optimization method and the numerical approximation technique used.

Did the authors incorporate their own version of MPC/AMPC, or did they simply use the MATLAB/Simulink toolbox?

Being purely modelling and simulation work, I expect a much more comprehensive analysis of the algorithm. 

Parametric robustness analysis, robustness analysis to external disturbances. Since MPC is very sensitive to model uncertainty, this is too important an aspect to be left out.

I also recommend using the Simulink Profiler to analyse computational complexity. Refer to the following works where the profiler is used to study the complexity of control algorithms for mechatronic systems:

https://www.mdpi.com/1996-1073/12/11/2224

This must necessarily lead to a discussion of possible implementation on an embedded system, with computational capabilities typical of an automotive ECU. That typically dedicates a Cortex-M4 type core to the control tasks.

Regarding the results presented:

since the "trajectory" track is mentioned in the title, I think it is mandatory to include the 2-D trajectory graph of the vehicle's centre of gravity.

The time series included is not very helpful in understanding the real behaviour resulting from the simulations.

It is asserted in the abstract that estimation of the wheel-road interaction model is critical at high speeds, however, the simulations are done at 60 km/h! This is inconsistent with the claims. 

I expect an analysis at varying speeds. Otherwise, one does not understand the value of including an algorithm that greatly increases the computational complexity required (which must be added to the MPC) if then the test speed is one for which even the linear wheel-road interaction model can give excellent results.

I hope the comments will help improve this work.

I honestly see a lot of effort for the authors. Good luck!

Reviewer 2 Report

The paper proposes an intelligent vehicle tracking control system.

The quality of the paper can be enhanced by addressing the following:

1- A table of acronyms would improve the readability of the paper.

2- The methodology section is missing

3- Figure 7 is not clear and should be regenerated.

4- The references are inadequate, and most are outdated. Authors are encouraged to add at least 7-10 recent references.

5- The conclusion is very short and should be revised to provide more about the study results.

6- I wonder what would be the effects of changing some parameters in the study on the performance of the proposed model. Authors are encouraged to address this.

7- In Figure 9a, I don't see much difference between the proposed model and others. What is the explanation for this?

Round 2

Reviewer 1 Report

I accept the author's responses, considering the actual version of the paper suitable for publication in the MDPI journal. 

Reviewer 2 Report

The authors have addressed all comments. The paper needs proofreading by a native speaker.
